# NMR Spectroscopy Identifies Chemicals in Cigarette Smoke Condensate That Impair Skeletal Muscle Mitochondrial Function

**DOI:** 10.3390/toxics10030140

**Published:** 2022-03-14

**Authors:** Ram B. Khattri, Trace Thome, Liam F. Fitzgerald, Stephanie E. Wohlgemuth, Russell T. Hepple, Terence E. Ryan

**Affiliations:** 1Department of Applied Physiology and Kinesiology, University of Florida, Gainesville, FL 32611, USA; rbk11@ufl.edu (R.B.K.); trthome@ufl.edu (T.T.); 2Department of Physical Therapy and Muscle Biology, University of Florida, Gainesville, FL 32611, USA; l.fitzgerald@phhp.ufl.edu (L.F.F.); rthepple@phhp.ufl.edu (R.T.H.); 3Department of Aging and Geriatric Research, University of Florida, Gainesville, FL 32611, USA; steffiw@ufl.edu; 4Center of Exercise Science, University of Florida, Gainesville, FL 32611, USA

**Keywords:** cigarette smoke extract, tobacco, mitochondria, chronic obstructive pulmonary disease, skeletal muscle, bioenergetics, smoking

## Abstract

Tobacco smoke-related diseases such as chronic obstructive pulmonary disease (COPD) are associated with high healthcare burden and mortality rates. Many COPD patients were reported to have muscle atrophy and weakness, with several studies suggesting intrinsic muscle mitochondrial impairment as a possible driver of this phenotype. Whereas much information has been learned about muscle pathology once a patient has COPD, little is known about how active tobacco smoking might impact skeletal muscle physiology or mitochondrial health. In this study, we examined the acute effects of cigarette smoke condensate (CSC) on muscle mitochondrial function and hypothesized that toxic chemicals present in CSC would impair mitochondrial respiratory function. Consistent with this hypothesis, we found that acute exposure of muscle mitochondria to CSC caused a dose-dependent decrease in skeletal muscle mitochondrial respiratory capacity. Next, we applied an analytical nuclear magnetic resonance (NMR)-based approach to identify 49 water-soluble and 12 lipid-soluble chemicals with high abundance in CSC. By using a chemical screening approach in the Seahorse XF96 analyzer, several CSC-chemicals, including nicotine, o-Cresol, phenylacetate, and decanoic acid, were found to impair ADP-stimulated respiration in murine muscle mitochondrial isolates significantly. Further to this, several chemicals, including nicotine, o-Cresol, quinoline, propylene glycol, myo-inositol, nitrosodimethylamine, niacinamide, decanoic acid, acrylonitrile, 2-naphthylamine, and arsenic acid, were found to significantly decrease the acceptor control ratio, an index of mitochondrial coupling efficiency.

## 1. Introduction

Approximately 25% of individuals with the tobacco smoke-related disease chronic obstructive pulmonary disease (COPD) have muscle atrophy and weakness, and this contributes to low mobility-related function, increased healthcare burden, and greater mortality [1,2,3]. The nature of muscle alterations is similar amongst different tobacco smoke-related diseases [4,5,6], including muscle atrophy and a shift towards more fatigue-prone fast muscle fibers, changes that are seen even in smokers who are free of tobacco smoke-related disease [7,8]. Mitochondrial impairments are well documented in skeletal muscle from COPD patients, including decreased oxidative capacity and elevated mitochondrial reactive oxygen species (ROS) [7,9,10]. Mitochondria are often described as the powerhouse of the cells, generating the energy required for many cell functions in the form of ATP by means of oxidative phosphorylation [11]. Both active and passive tobacco/cigarette smoke exposure decreases mitochondrial respiration and, in some cases, increases ROS across a range of cell/tissue types (nicely reviewed by Fetterman et al. [12]). Interestingly, there are contradicting reports in the literature regarding whether short-term tobacco smoke exposure negatively impacts muscle mitochondrial function in mice [13,14]. A potential explanation for this discrepancy could be that one study employed in vivo (magnetic resonance-based) methods, whereas the other study examined mitochondria ex vivo in a respirometer where oxygen delivery is not a limiting factor. Thus, the degree to which acute tobacco smoke exposure can suppress mitochondrial respiratory function is unclear.

Importantly, COPD is considered to be a chronic inflammatory lung disease that is typically caused by long-term exposure to gaseous irritants or particulate matter, commonly related to cigarette/tobacco smoking. As such, the studies above documenting muscle/mitochondrial abnormalities in COPD patients primarily enrolled former smokers rather than those actively exposed to cigarette smoke. Thus, far less is known about the acute effects of cigarette/tobacco smoke on skeletal muscle biology. However, there are reports suggesting that active exposure to tobacco smoke has direct/acute impacts on muscle health and function. For example, Darabseh et al. [15] subjected active cigarette smokers to a 14-day smoking cessation protocol and observed an improvement in muscle fatigue resistance and lowered biomarkers of inflammation. In mice, ~12 weeks of cigarette smoke exposure was reported to decrease muscle mitochondrial respiration, which was reversed following 2-weeks of smoking cessation [16]. Furthermore, the same study demonstrated that acutely treating healthy soleus muscles with cigarette smoke extract impaired mitochondrial respiration [16]. Additional work in rats reported a reduction in muscle citrate synthase activity, a marker of mitochondrial content, following only 7-days of cigarette smoke exposure [17]. Therefore, these studies demonstrate that active smoking acutely impairs muscle mitochondrial health, which can be recovered upon smoking cessation, suggesting that tobacco smoke contains chemicals that are toxic to muscle mitochondria.

Tobacco smoke consists of gaseous suspended droplets, which can contain over 8700 chemical constituents [18,19,20,21,22,23,24,25,26,27,28,29,30,31,32,33,34,35,36]. Considering this complexity, it is likely that tobacco smoke contains many chemical constituents that could directly impair mitochondrial energy transduction. Consistent with this notion, exposure to tobacco smoke negatively alters the mitochondrial structure and/or function across a range of cell types [37,38,39], even with acute exposures or treatments. However, the underlying biochemical mechanisms driving impaired mitochondrial function with tobacco smoke exposure is unclear, and few studies have explored skeletal muscle mitochondria specifically. In order to address this gap in knowledge, we employed an analytical chemistry approach utilizing nuclear magnetic resonance (NMR) [40] to first identify chemical constituents in cigarette smoke condensate (CSC). Once identified, we performed a chemical screen of the most abundant chemicals detected in CSC to uncover individual chemicals that negatively impact muscle mitochondrial respiratory function using a mouse model.

## 2. Materials and Methods

### 2.1. Chemicals

Deuterium oxide (D_2_O) and deuterated chloroform were purchased from Cambridge Isotope Laboratories, MA, USA. Deuterated 4,4-dimethyl-4-silapentane-1-sulfonic acid (DSS) was obtained from Fujifilm Wako pure chemical Corporation, VA, USA. Sodium azide (NaN_3_), sodium monophosphate and diphosphate, and all the chemicals listed in Appendix A were procured from Millipore-Sigma, MO, USA. The remaining chemicals and reagents were purchased from either GIBCO (ThermoFisher, Waltham, MA, USA), Research Products International, Combi blocks, VWR Suppliers, or Millipore Sigma, as described in our previous work [41]. We obtained CSC from Murty Pharmaceuticals, which was prepared by smoking University of Kentucky’s 3R4F Standard Research Cigarettes on an FTC Smoke Machine. The Total Particulate Matter (TPM) on the filter was calculated by the weight added to the filter after smoking. From the TPM, the amount of DMSO was calculated to allow extraction of a 4% (40 mg/mL) solution. The condensate is extracted with DMSO by soaking and sonication.

### 2.2. Animals

Male and female C57BL/6J mice (Stock #000664) were obtained from The Jackson Laboratory at 4-months of age. All mice were housed in a temperature (22 °C) and light-controlled room (12-h light/12-h dark) and were maintained on a standard chow diet (Envigo Teklad Global 18% Protein Rodent Diet 2918 irradiated pellet) with ad libitum access to food and water. All animal experiments adhered to the Guide for the Care and Use of Laboratory Animals from the Institute for Laboratory Animal Research, National Research Council, Washington, D.C., National Academy Press, 1996, and any updates. All procedures were approved by the Institutional Animal Care and Use Committee of the University of Florida. For mitochondria isolation, all mice were sacrificed between 9 a.m. and 10 a.m. eastern standard time to account for circadian influences on metabolism.

### 2.3. Isolation of Skeletal Muscle Mitochondria

Isolation of skeletal muscle mitochondria was performed following protocols described previously [41,42,43]. In short, dissection of skeletal muscle was performed after cervical dislocation of mice, followed by trimming to remove fat, tendon, and connective tissues. Next, the muscle tissue was minced and subjected to 5-min trypsin digestion on ice and subsequently centrifuged at 200× *g* for 10 min at 4 °C to remove trypsin. The pellet was resuspended with ice-cold mitochondrial isolation medium (MOPS (50 mM), KCl (100 mM), EGTA (1 mM), MgSO_4_ (5 mM), pH = 7.1) supplemented with 2 g/L bovine serum albumin (BSA), homogenized via a glass Teflon homogenizer (Wheaton), and immediately centrifuged for 10 min maintaining 800× *g* at 4 °C. The supernatant portion was collected and centrifuged at 10,000× *g* to pellet mitochondria. The resulting mitochondria were gently resuspended in 200 µL of mitochondrial isolation buffer with no BSA, and protein concentration was determined via a bicinchoninic acid protein assay (ThermoFisher #A53225).

### 2.4. Seahorse XFe96 Assay: Oxygen Consumption by Skeletal Muscle Mitochondria

The Seahorse XFe96 Analyzer (Agilent) was used to perform oxygen consumption rate (OCR) measurements from mitochondria isolated from skeletal muscle. The Seahorse XFe96 96-well cartridges were hydrated overnight and switched to the Seahorse XF Calibrant (pH 7.4) solution the following morning. The cartridge was loaded with the desired concentration of either CSC or the individual chemicals, as well as mitochondrial substrates/inhibitors (ADP, succinate, rotenone). Five micrograms of mitochondria were seeded in 50 µL of mitochondrial assay buffer (MAB) supplemented with 5 mM pyruvate and 2.5 mM malate, using Agilent Seahorse XF96 cell culture microplates. The MAB consisted of: 5 mM magnesium chloride (MgCl_2_), 105 mM MES potassium salt, 30 mM potassium chloride (KCl), 10 mM potassium biphosphate (KH_2_PO_4_), 2.5 g/L of bovine serum albumin (BSA), 1 mM ethylene glycol-bis(β-aminoethyl ether)-N,N,N′,N′-tetra-acetic acid (EGTA), and 20 mM creatine with pH 7.2. The 96-well microplate was centrifuged at 2250× *g* for 20 min, maintaining a temperature of 4 °C to ascertain homogenous adherence of cells at the bottom of each well. MAB was also utilized to prepare CSC (0.02%, 0.1 %, and 1%), 0.1% chemicals, 1 mM (final) adenosine 5′-diphosphate (ADP), 5 mM (final) succinate, and 50 µM (final) rotenone to be injected. In the case of water-insoluble chemicals, these were first dissolved in dimethyl sulfoxide (DMSO), and the MAB buffer was used for dilution.

### 2.5. NMR Sample Preparation

No extraction was performed on CSC samples. Two sets of NMR samples were prepared for each solvent system used. For one set of samples, a lyophilizer (Labconco Freezone 2.5 L, Kansas, MO, USA) was used to dry out DMSO present in the 45 µL of CSC sample. The dried sample was dissolved in 100% deuterated water consisting 50 mM phosphate buffer (pH 7.2) along with 0.5 mM DSS, 0.2% NaN_3_, and 2 mM EDTA. Another sample was prepared by adding 10% volume:volume (*v*/*v*) of deuterated water, 0.2% NaN_3_, and 0.5 mM DSS (final concentration) to the un-dried 45 µL CSC aliquot to identify the volatile chemicals. Similarly, the lipid-soluble compounds were monitored by dissolving the CSC sample in CDCl_3_ with 10 mM of pyrazine (as internal standard). The dried CSC powder obtained after lyophilization was resuspended in 70 µL CDCl_3_ consisting 10 mM of pyrazine. In order to track down volatile lipid-soluble compounds, 45 µL of the un-dried CSC sample was placed in a glass vial, and 30 µL (40% *v*/*v*) of CDCl_3_ was added to it. All samples were loaded into 1.7 mm O.D. NMR tube (CortecNet Corp, Brooklyn, NY, USA) for NMR experiments acquisition. One- and two-dimensional NMR spectra were acquired using a CP TXI CryoProbe with an Avance II Console (14.1 T, Bruker Biospin, Billerca, MA, USA) NMR instrument at Advanced Magnetic Resonance facility at McKnight Brain Institute, the University of Florida. The first slice of a nuclear Overhauser effect spectroscopy (NOESY) pulse sequence (tnnoesy) [44] was used with parameters previously described [45,46,47,48]. Five hundred and twelve scans were collected for 1D NOESY spectra. Two-dimensional spectra including heteronuclear single quantum coherence (HSQC) [49], correlated spectroscopy (COSY) [50,51], and total correlated spectroscopy (TOCSY) [52] were also acquired to validate chemical assignments. HSQC NMR spectrum was collected with relaxation delay (d1) of 1.5 s, 24 scans (nt), spectral width (sw) of 7142.9 and 33,112.6 Hz in F2 and F1 dimensions, respectively. A 90-degree pulse width (pw) was used with 0.14 s acquisition time (at) with GARP4 ^13^C decoupling. For COSY and TOCSY NMR, d1 of 1.5 s, sw of 7142.9 Hz in both F2 and F1 dimensions, a 90-degree pulse, and acquisition time of 0.1434 s were used. The nt used for COSY was 16, and for TOCYS, it was 32. All the 1D and 2D experiments were acquired at room temperature (25 °C).

### 2.6. Data Processing

MestReNova (version 14.1.2-25024; Mestrelab Research, S.L., Santiago de Compostela, Spain) was utilized to process NMR spectra. All 1D spectra were subjected to exponential line broadening of 0.22 ppm before Fourier-transformation. Furthermore, Spline baseline and phase corrections were applied. All 2D spectra were zero-filled (2048) before Fourier-transformation. Phase correction (except COSY), T1-noise reduction, and Whittaker Smoother baseline correction were applied on both dimensions. All aqueous-phase spectra were calibrated with a DSS peak at 0 ppm. The lipid phase spectra were calibrated with CDCl_3_ resonance at 7.26 ppm. Chenomx nmr suite 8.43, Alberta, Canada, was used to assign and quantitate most of the water-soluble compounds. A few water-soluble chemicals and all lipid-soluble chemicals integrated peak areas were used to calculate concentrations. For the aqueous phase compounds that were not present in the Chenomx nmr suite 8.6 and all lipid-soluble compounds, we used several resources, including the Biological Magnetic Resonance Data Bank (BMRB) [53], Human Metabolome Database (HMDB) [54], a set of 2D experiments (Appendix A), as well as published reports [35,36,55,56,57,58,59,60,61,62,63,64] to assist with chemical identification. In addition, spiking ^1^H 1D NOESY NMR experiments were carried out for these chemicals to validate their presence in CSC.

### 2.7. Statistical Analysis

All data are presented in means ± S.D. format. The normality of data was confirmed using the Shapiro–Wilk test. Significance for the data involving multiple groups was analyzed using either a two-way or one-way ANOVA with correction for multiple comparisons using the Original FDR method of Benjamini and Hochberg via GraphPad Prism (version 9.0.2 (121), GraphPad Software, San Diego, CA, USA), considering *p* < 0.05 as statistically significant.

## 3. Results

### 3.1. Acute Treatment with CSC Impairs Complex 1 Dependent Respiration in Skeletal Muscle Mitochondria

Smoking is a major risk factor for many chronic diseases, and some of the chemicals derived from tobacco smoke are known to be toxic [65]. To begin to understand how these toxic chemicals impact mitochondrial function in skeletal muscle, we obtained CSC derived from research cigarettes (3R4F) to use in experiments of mitochondrial respiratory function in vitro following acute direct exposures to CSC. The overall schematic diagram for the workflow employed in this study is shown in Figure 1.

First, we examined the dose-dependent effect of three different concentrations (0.02%, 0.1%, and 1%) of CSC on skeletal muscle mitochondrial respiratory function in mitochondrial isolates obtained from C57BL6J male mice (*n* = 3 biological samples). Freshly isolated muscle mitochondria were seeded into Seahorse XF 96 V3 PS cell culture microplates and were fueled with pyruvate and malate (State 2 respiration), followed by the addition of CSC for an acute incubation (~10 min). Next, State 3 respiration was initiated by the addition of ADP, followed by subsequent additions of succinate and rotenone via separate injections. This protocol allowed for the sequential measurement of baseline State 2 respiration, Complex I ADP stimulated respiration (State 3), Complex I + II respiration stimulated by the addition of succinate, and Complex II respiration only following the addition of the Complex I inhibitor, rotenone. Compared to DMSO-treated control mitochondria, a dose-dependent decrease in Complex I ADP stimulated respiration was observed with CSC (Figure 2A–C). Notably, 1% CSC-treated mitochondria were unable to increase Complex I respiration following the addition of ADP, whereas treatment at 0.1% and 0.02% reduced this rate by 70% and 51%, respectively. However, 1% CSC-treated mitochondria displayed an ability to increase respiration following the addition of succinate, although the maximal Complex II respiratory rate was still impaired by 60% compared to DMSO control-treated mitochondria. At 0.1% CSC concentration, ADP-induced Complex I respiration was reduced by 70% when compared to DMSO control. Similarly, we observed a ~17% reduction in State 2 respiration following 0.1% CSC exposure and ~12% reduction in succinate stimulated respiration through mitochondrial Complex II. Due to the nature of the experiment (direct short-term exposure of CSC to mitochondria), these results demonstrate that CSC acutely impairs respiration in mitochondria isolated from mouse skeletal muscle.

### 3.2. Detection of Water and Lipid Soluble Chemicals in 3R4F-Derived CSC via 1D/2D NMR

Next, we employed 1D and 2D NMR approaches to identify individual chemicals present in the CSC. NMR spectroscopy is a well-established analytical tool for analyzing chemicals present in a wide variety of samples that include biological fluids, foods, beverages, and others [36]. Extensive public NMR data sets were developed and can be utilized to identify the chemical structures observed in NMR spectra. Our NMR analyses identified 49 highly concentrated water-soluble chemicals in CSC (as shown in the ^1^H NMR spectra in Figure 3 and listed in Appendix A). Non-volatile water-soluble chemicals were determined using a high resolution ^1^H NMR spectrum of dried CSC powder (obtained after drying out 45 µL of 4% CSC) dissolved in 50 µL of 50 mM phosphate buffer (pH 7.2) with 0.5 mM DSS in a deuterated environment (Figure 3A). Volatile water-soluble chemicals were also identified and annotated (Figure 3B) using 45 µL of 4 % CSC solution with 5 µL Chenomx standard (consisting of 5 mM of DSS with 0.02% NaN_3_ in D_2_O). For lipid-soluble chemicals, dried powder of 4% CSC was dissolved in CDCl_3_ containing 10 mM of pyrazine (as an internal reference). A total of 12 lipid-soluble chemicals were identified and annotated (Appendix A). The measured concentration of water-soluble chemicals was determined with respect to DSS peak at 0.00 ppm either via Chenomx NMR Suite 8.6 or using the integrated peak area. For lipid-soluble chemicals, integrated peak areas of the respected chemicals and pyrazine peak at 8.61 ppm were used to determine the concentration. Some chemicals such as glycerol (20.51 mM), quinolone (9.22 mM), acetone (4.68 mM), ethylene glycol (2.12 mM), acetonitrile (7.88 mM), triacetin (1.33 mM), fatty acids (9 mM; a mixture of several fatty acids including decanoic acid as shown in Appendix A), and pyrogallol (1.07 mM) were also detected.

### 3.3. Mitochondrial Screening Identifies Nicotine, Decanoic Acid and o-Cresol as Toxins That Impair Complex I-Supported Mitochondrial Respiration

Of the identified CSC chemicals with relatively high concentration (≥0.02 mM; excluding few amino acids, sugars, and common solvents), 29 chemicals were screened for their impact on mitochondrial respiratory function (shown in Table 1). Five additional chemicals in CSC that were previously reported to impair mitochondrial function in non-muscle cells [36,66,67] were also screened (even though they were undetectable by 1D/2D NMR in this study) (last five chemicals of Table 1). Because 0.1% CSC was found to impair respiration in isolated mitochondria from skeletal muscles significantly, we screened each individual CSC chemical at the concentration it is present in 0.1% CSC. Isolated skeletal muscle mitochondria were obtained from both male and female adult C57BL6J mice and energized with pyruvate/malate, followed by acute exposures to each chemical and assessment of mitochondrial oxygen consumption using the Seahorse XF96 Analyzer with the bioenergetic assessment protocol described in Figure 4A. State 2 (pyruvate/malate in the absence of ADP) respiration was significantly increased by several CSC chemicals including propylene glycol (125.54 ± 28.38, *p* = 0.044), α-pinene (125.01 ± 21.97, *p* = 0.041), decanoic acid (128.44 ± 35.40, *p* = 0.02), acrylonitrile (138.72 ± 44.19, *p* = 0.002), and arsenic acid (143.18 ± 35.33, *p* = 0.0005) (Figure 4B).

Surprisingly, only three CSC chemicals, nicotine (83.90 ± 6.57, *p* = 0.0008), decanoic acid (83.80 ± 6.76, *p* = 0.008), and o-Cresol (87.28 ± 5.83, *p* = 0.0008), were found to impair Complex I-supported State 3 respiration (Figure 4C). Following the addition of succinate to additionally fuel Complex II of the electron transport system, only nicotine (88.68 ± 7.28, *p* = 0.015) and o-Cresol (90.02 ± 7.06, *p* = 0.0261) significantly impaired succinate-stimulated Complex I +II State 3 respiration (Figure 4D), whereas the effect of decanoic acid did not reach statistical significance (92.47 ± 7.16, *p* = 0.0925). Consistent with observations of CSC treatment, none of these three chemicals were found to decrease Complex II State 3 respiration. However, only phenylacetate was found to significantly impair State 3 respiration supported only by the Complex II substrate succinate (88.06 ± 8.39, *p* = 0.040), highlighting the diversity with which tobacco smoke chemicals can impact mitochondrial respiration (Figure 4E).

Several chemicals tested has non-significant changes in State 2 and Complex I-supported State 3 (Figure 4B,C) which may have biological relevance to OXPHOS coupling efficiency. Thus, we calculated the acceptor control ratio (ACR, State 3 divided by State 2) for all chemicals (Figure 4F). A decrease in ACR may reflect either impairments in OXPHOS or increases in proton leak, both of which are largely considered to have negative impacts on energy transduction. Several chemicals were found to significantly decrease the ACR compared to DMSO-treated control mitochondrial including nicotine (*p* = 0.0099), o-Cresol (*p* = 0.0358), decanoic acid (*p* = 0.0011), quinoline (*p* = 0.0368), propylene glycol (*p* = 0.0051), myo-inositol (*p* = 0.007), nitrosodimethylamine (*p* = 0.0302), niacinamide (*p* = 0.0302), acrylonitrile (*p* = 0.0311), 2-naphthylamine (*p* = 0.0119), and arsenic acid (*p* = 0.001).

## 4. Discussion

Patients with diseases that stem from chronic tobacco smoke exposure, such as COPD, exhibit muscle pathology, including atrophy, weakness, and decreased exercise performance that contributes to higher morbidity and mortality risk [6,8,65]. COPD patients were reported to have decreased skeletal muscle oxidative function using both in vivo [10,68,69] and ex vivo assessments [70,71,72,73,74]. Whereas COPD is often considered as a consequence of long-term smoking, several studies reported that acute cigarette smoke exposure can impair muscle mitochondrial function [16,17] and that short-term smoking cessation can reverse these effects [15,16], suggesting the presence of acute muscle mitochondrial toxicity with smoking. In the present study, we first investigated the effect of CSC on mitochondrial respiratory function and found that acute exposure of muscle mitochondria to CSC resulted in a dose-dependent impairment in mitochondrial respiration (Figure 2). Next, we used NMR spectroscopy to profile the chemical composition of CSC, and following chemical identification, we screened a subset of chemicals (in high concentration) for their ability to impact skeletal muscle mitochondrial respiration following acute exposure negatively. Screening of select chemical components of CSC identified nicotine, decanoic acid, and o-Cresol as chemicals that impair mitochondrial respiratory capacity, particularly when mitochondria were fueled with the Complex I substrates, pyruvate and malate. Interestingly, only phenylacetate displayed a mild (~7%) impairment in mitochondrial respiration when fueled by succinate (Complex II substrate in the presence of rotenone).

State 2 respiration, which is often described as LEAK respiration, was significantly elevated by propylene glycol, α-pinene, decanoic acid, acrylonitrile, and arsenic acid. LEAK respiration is generally attributed to several processes not linked with ATP production, including proton leak, substrate transport, cation cycling, and in some cases, ROS production [75]. Propylene glycol is a commonly used food additive and a vehicle for several types of pharmaceutical preparations. It can be metabolized by several alcohol dehydrogenases to form lactate and pyruvate, which can serve as fuel for mitochondria. Alpha-pinene is an organic terpene consisting of two isoprene units and is found in the oils of many trees/plants. Because of its structure, alpha-pinene is lipophilic and could possibly accumulate in mitochondrial membranes and thereby alter energetics. While direct support for this notion is not available, alpha-pinene was shown to reduce pathology in ischemic stroke models [76], which is known to be mediated in part by mitochondrial ROS, which is highly dependent on the mitochondrial membrane potential. Acrylonitrile is a well-known toxin in cigarette smoke and was found herein to elevate State 2, or LEAK respiration in muscle mitochondria. This observation could stem from its conjugation to glutathione, which could have increased ROS levels, or the fact that one of its metabolites is cyanide, a known inhibitor of mitochondrial Complex IV. Similarly, arsenic acid (arsenic chloride) was revealed to increase the ROS level of mitochondria [77] which could explain the increased State 2 respiration observed with acute exposure in this study.

Nicotine is a natural alkaloid and a principal addictive constituent in tobacco and is well known for its psychopharmacological effects [78]. Our results indicate that nicotine alone significantly impairs ADP-stimulated skeletal muscle mitochondrial respiration supported by pyruvate/malate alone and following the addition of succinate (Figure 4). Interestingly, upon inhibition of Complex I with rotenone (leaving mitochondrial to respire with Complex II substrates only), the detrimental impact of nicotine alone was abolished, suggesting its mechanism of action involves either NADH dehydrogenase (Complex I) or other matrix dehydrogenase enzymes. This observation is consistent with previous studies reporting nicotine-mediated mitochondrial respiration impairment in non-muscle tissues [78,79,80,81]. For example, treatment with 1 mg/kg/day of nicotine for seven days impaired brain mitochondrial function, including Complex I activity, in Wistar rats [81]. This observation is consistent with the results of Cormier et al. [78], who reported that nicotine exhibits inhibitory effects on Complex I of the mitochondrial electron transport system in rat brains. Further to this point and consistent with our results, Dewar et al. reported that nicotine (50 µM to 1.25 mM) did not impact succinate-supported respiration in mitochondria isolated from the liver [82]. The mechanism underlying the impairment in mitochondrial respiration caused by nicotine is not fully known; however, one plausible explanation appears to involve direct inhibition of NADH-dehydrogenase (Complex I of the mitochondrial electron transport system) [79,80].

o-Cresol is a methyl-substituted phenol at the ortho position and is found in cigarette smoke [83,84]. It is considered toxic and has been associated with acute respiratory distress syndrome along with cardiovascular, renal, and hepatic pathology following inhalation exposures [85]. In the current study, o-Cresol significantly impaired ADP-stimulated skeletal muscle mitochondrial respiration fueled by pyruvate/malate alone, as well as pyruvate/malate and succinate together. This result is consistent with a previous report [84] demonstrating that o-Cresol, along with its meta- and para-isomers, reduced NAD-linked and succinate-linked respiration in rat liver mitochondria, with the former being more affected. Cresol isomers, including o-Cresol, are also used as active ingredients in bactericides and some disinfectants where their mechanism of action involves disruption of bacterial cell membranes. Disruption of the mitochondrial membranes would dissipate the proton motive force, the driving force controlling mitochondrial oxidative phosphorylation, and thus could explain the observed results. Interestingly, phenylacetate, an ester form of phenol and acetic acid, was found to negatively impact muscle mitochondrial respiration when supported only by the Complex II substrate succinate. In addition to being a chemical in CSC, the elevation of phenylacetate was observed in diseases such as sepsis [86], botulinum [87], and end-stage renal disease [86,88]. In accordance with our results, phenylacetate inhibits liver mitochondrial respiration and lowers the calcium retention capacity—an indicator of increased susceptibility to mitochondrial permeability transition pore opening [86]. In smooth muscle cells, phenylacetate also increases ROS production [86,88]. Future work is needed to determine the biochemical mechanisms underlying the detrimental effects of these chemicals on mitochondrial energy transduction.

Another interesting finding was that CSC contains a substantial amount of fatty acid derivatives. One fatty acid, decanoic acid, was detected at relatively high concentrations (~9 mM) in CSC and was found to significantly impair the ADP-induced Complex I skeletal muscle mitochondrial respiration without affecting succinate-induced Complex II respiration. Consistent with our data, several previous papers reported that different classes of long-chain free fatty acids exhibit inhibitory effects on mitochondrial Complex I of the ETS, with no significant effect on succinate-supported respiration [89,90,91,92,93,94]. Similarly, decanoic acid was reported to alter Complex I respiration in the mitochondria of neuronal cell lines [95], and tetradecanoic acid (a larger analog of decanoic acid) was shown to block electron transfer in between NADH dehydrogenase and ubiquinone of the ETS in bovine heart submitochondrial particles [94]. Similarly, hexadecanoic acid (palmitic acid) also impairs NADH dehydrogenase (Complex I) activity [89]. One proposed explanation for these effects involves a hydrophobic cavity present in an iron–sulfur protein of Complex 1 that can serve as a binding site for hexadecanoic acid (or similar free fatty acids), which subsequently interferes with electron transfer [96,97].

Whereas only nicotine, o-Cresol, and decanoic significantly impaired State 3 mitochondrial respiration supported by Complex I substrates, eight additional chemicals were found to significantly decrease the ACR in muscle mitochondria (Figure 4F). The ACR, calculated by dividing the Complex I State 3 respiration by State 2 respiration, is an index of OXPHOS coupling efficiency and is commonly used as a general indicator of mitochondrial function [98]. While these eight additional chemicals (quinoline, propylene glycol, myo-inositol, nitrosodimethylamine, niacinamide, acrylonitrile, 2-naphthylamine, and arsenic acid) did not have statistically significant changes in either State 2 or State 3 respiration, non-significant changes in one or both variables used to determine ACR likely indicate modest levels of mitochondrial dysfunction caused by these chemicals. Future work is needed to determine if these chemicals exert their effects by impairing oxidative photophosphorylation or by enhancing proton leak.

There are some limitations of the present study worthy of discussion. First, there are an estimated 8700 chemicals reported to be present in tobacco smoke [18,99,100], and the current study has only screened 34 chemicals identified as highly concentrated. Moreover, there are several tobacco smoke chemicals that are present in the gaseous state, are highly volatile, or lack protons, and thus are not detectable by NMR [18,21,99,101,102,103,104,105,106]. Second, there are likely toxic chemicals in CSC found within the low µM or nM concentration that were not detectable because of the sensitivity of NMR. Using MS or chromatography methods [107,108,109,110,111] is another option for chemical identification; however, this approach requires an extensive chemical library for definitive identification through standard curves. Third, this study examined the acute impact of CSC chemicals on mitochondrial respiratory function. The possibility that chronic exposure to some chemicals could alter cellular signaling pathways, leading to a cumulative pathological effect that manifests as impaired muscle mitochondrial function, is yet to be explored. Additionally, although electronic tobacco products have increased in popularity over recent years as an alternative for conventional cigarettes, the relevance of these findings to electronic tobacco products is not known at this time because of potential differences in the chemical constituents generated by electronic versus traditional tobacco cigarettes. Fourth, to our knowledge, there are no studies documenting the level of smoke-derived toxic chemicals in skeletal muscle. Thus, extrapolations of findings from acute exposures of CSC doses ranging from 0.02 to 1.00% *v*/*v* to in vivo muscle physiology are cautioned.

## 5. Conclusions

This study found that skeletal muscle mitochondrial respiration was dose-dependently impaired by acute exposure to cigarette smoke extract. By using NMR-based approaches, this study analyzed the chemical composition of cigarette smoke extract and subsequently performed a chemical screening analysis to identify individual chemicals contributing to the observed pathologic effects in mitochondrial energetics. Several chemicals, including nicotine, o-Cresol, decanoic acid, and phenylacetate, were found to impair ADP-stimulated respiration modestly, and eight additional chemicals decreased the ACR in mitochondrial isolates prepared from murine skeletal muscle. These findings add biochemical resolution to the existing literature documenting muscle mitochondrial abnormalities in conditions stemming from acute tobacco smoke exposure.

## Figures and Tables

**Figure 1 toxics-10-00140-f001:**
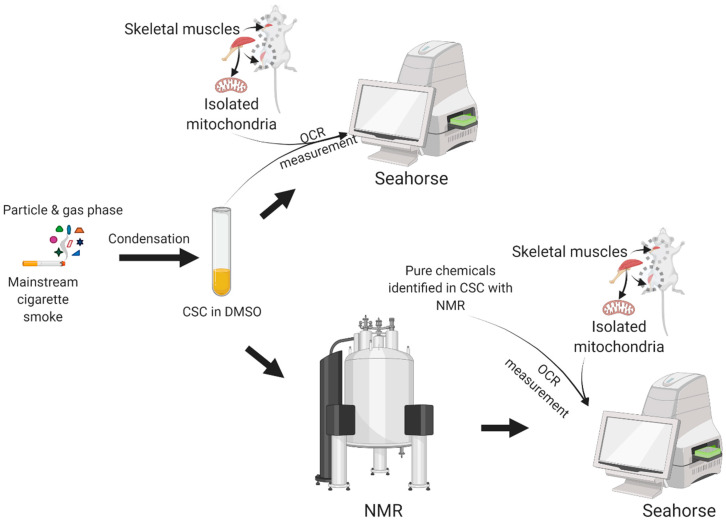
Schematic diagram for the overall workflow employed in this study including CSC sample generation, tissue collection, respirometric analysis using Seahorse instrument, and NMR screening of CSC to identify the chemicals present in CSC. OCR: oxygen consumption rate, NMR: nuclear magnetic resonance, CSC: cigarette smoke condensate.

**Figure 2 toxics-10-00140-f002:**
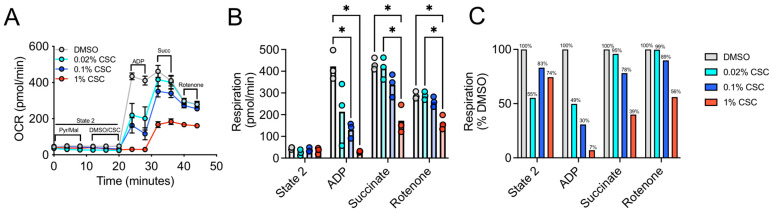
Impact of acute CSC treatment on mitochondrial respiratory function. (**A**) Seahorse analysis of mitochondrial respiration indicating addition of drugs/substrates/inhibitors. (**B**) Quantification of respiration based on conditions for each dose of CSC. (**C**) Changes in respiration presented as a percentage of the DMSO control group. (*n* = 3/group) Two-way ANOVA (including Tukey’s post hoc testing) was conducted via GraphPad Prism (version 9.0.2), * *p* < 0.05 as statistically significant. OCR: oxygen consumption rate, CSC: cigarette smoke condensate, DMSO: dimethyl sulfoxide, Pyr: pyruvate, Mal: malate, ADP: adenosine diphosphate, Succ: succinate.

**Figure 3 toxics-10-00140-f003:**
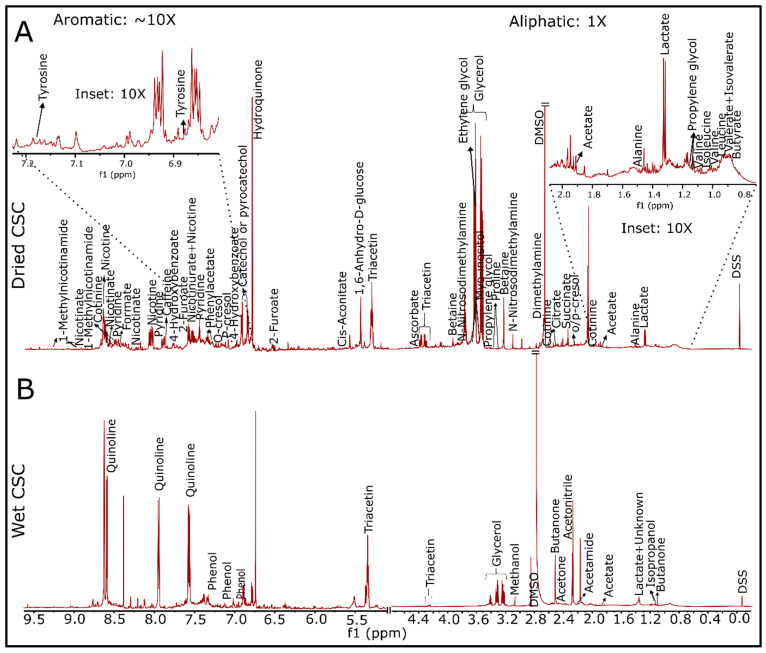
^1^H NMR spectra showing the water-soluble chemicals in 4% cigarette smoke condensate. A total of 49 chemicals were identified and annotated (listed in Appendix A) with the help of Chenomx NMR Suite 8.6, biological magnetic resonance bank (BMRB), and several published literatures. The upper spectrum showed all non-volatile water-soluble chemicals, while the bottom spectrum showed some volatile water-soluble chemicals. CSC: cigarette smoke condensate, DMSO: dimethyl sulfoxide.

**Figure 4 toxics-10-00140-f004:**
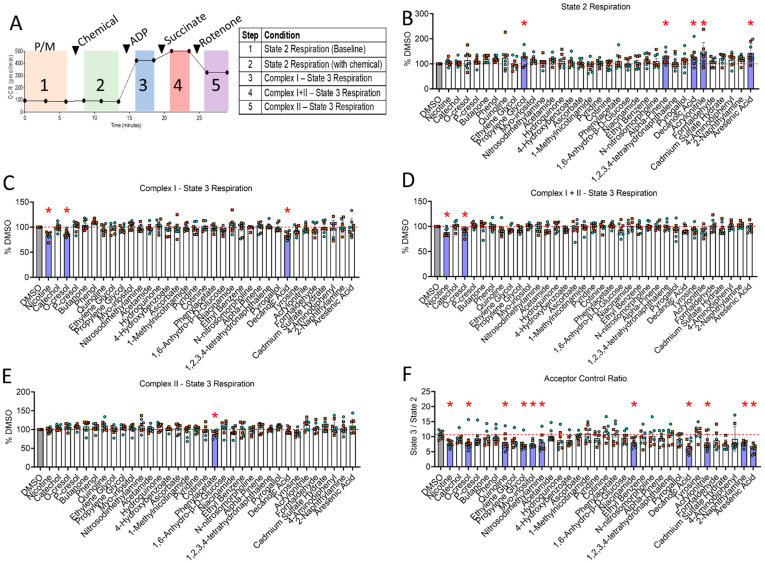
(**A**) Graphical representation for OCR levels given by mitochondrial respiration function parameters (y-axis) versus time (in min, x-axis) is shown with each sequential substrate/inhibitor addition. (**B**) State 2 respiration with mitochondria was energized with pyruvate/malate in the absence of exogenous ADP. (**C**) Quantification for maximal State 3 (ADP stimulated) respiration through mitochondrial Complex I (pyruvate/malate + ADP). (**D**) Quantification for maximal State 3 (ADP stimulated) respiration through mitochondrial Complex I + II (pyruvate/malate/succinate + ADP) (**E**) Quantification for maximal State 3 (ADP stimulated) respiration through mitochondrial Complex II only after inhibition of Complex I with rotenone, respectively. (**F**) The acceptor control ratio (ACR) calculated by dividing State 3 by State 2 respiration supported by Complex I. *n* = 4 animals per sex. Teal symbols represent male mice; orange symbols represent female mice. * *p* < 0.05 vs. DMSO using one-way ANOVA and FDR-correction for multiple comparisons. Abbreviations: P/M: pyruvate/malate; ADP: adenosine diphosphate; OCR: oxygen consumption rate; P/M: pyruvate/malate; DMSO: dimethyl sulfoxide; ADP: adenosine diphosphate.

**Table 1 toxics-10-00140-t001:** Chemicals detected in CSC and their concentrations that were screened for impact on muscle mitochondrial respiration.

S.No.	Chemicals	Concentration (mM)	S.No.	Chemicals	Concentration (mM)
1	Quinoline	9.22	18	Propylene glycol	0.14
2	Decanoic acid	9.0	19	o-Cresol	0.087
3	Ethylene glycol	2.12	20	Ethyl benzene	0.08
4	Butanone	1.46	21	N-nitrosomorpholine	0.07
5	Pyrogallol	1.07	22	p-cresol	0.06
6	Phenol	0.73	23	Formaldehyde	0.06
7	Acetamide	0.69	24	Tyrosine	0.04
8	Myo-Inositol	0.68	25	Pyridine	0.04
9	Nicotine	0.65	26	Phenylacetate	0.04
10	1,2,3,4-tetrahydronaphthalene	0.64	27	4-Hydroxybenzoate	0.03
11	N-Nitrosodimethylamine	0.59	28	Niacinamide	0.02
12	1,6-Anhydro-β-D-glucose	0.47	29	1-Methylnicotinamide	0.007
13	Hydroquinone	0.43	30	Acrylonitrile	0.76
14	Catechol	0.37	31	2-Naphthylamine	0.0042
15	α-pinene	0.34	32	Arsenic (III) chloride	0.0016
16	Ascorbate	0.28	33	Cadmium sulfate hydrate	0.0016
17	Cotinine	0.183	34	4-Aminobiphenyl	0.000061

## Data Availability

All data acquired and analyzed for this study can be found in Appendix A provided with this manuscript.

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
