# Peer review of "NMR Spectroscopy Identifies Chemicals in Cigarette Smoke Condensate That Impair Skeletal Muscle Mitochondrial Function"

_toxics, 2022, doi:10.3390/toxics10030140_

Round 1
Reviewer 1 Report
Journal: Toxics
Manuscript number #:
Article type: Research Article
Title: NMR spectroscopy identifies chemicals in cigarette smoke condensate that impair skeletal muscle mitochondrial function
The authors investigated whether cigarette smoke condensate (CSC) acutely reduces respiration capacity of muscle mitochondria. Chemical substances contained in CSC were individually identified using NMR-based approaches, and which substances impair skeletal muscle mitochondrial function were identified. This study deals with an interesting topic, however, there are some points to be improved as follows.
Major comments:
- Figure 3 does not include Figure 3A and Figure 3B separately, but Figure 3A and Figure 3B are described in the Results section. To avoid the reader confusion, this needs to be clarified and corrected.
- Authors need to describe in detail the limitations and supplementary points of this study in the Discussion section.
Minor comments:
- The full name of the abbreviation should be written only once, and after that, only the abbreviation should be written in the manuscript. Some full names are mentioned more than once (e.g., cigarette smoke condensate (CSC)).
- The right side of the Figure 2C was cropped. Please check and correct
- There are some typos in this manuscript. Please check and correct.
Decision: Minor revision
Reviewer 2 Report
In this study, the authors found that skeletal muscle mitochondrial respiration was dose-dependently impaired by acute exposure to cigarette smoke extract(CSC). Then they analyzed the chemical composition of cigarette smoke extract and subsequently performed a chemical screening analysis to identify individual chemicals contributing to the observed pathologic effects in mitochondrial energetics by NMR-based approaches. Several chemicals (nicotine, o-cresol, decanoic acid, and phenylacetate) were found to modestly impair ADP-stimulated respiration and eight additional chemical decreased the ACR in mitochondrial isolates prepared from murine skeletal muscle. This work is novel and I recommend publication after addressing the following points.
Comments:
- First, you examined the dose-dependent effect of three different concentrations (0.02%, 0.1%, and 1%) of CSC on skeletal muscle mitochondrial respiratory function in mitochondrial isolates obtained from C57BL6J mice (line 100-102). Why did you choose these concentrations? What are the research implications of these concentration Settings?
- Notably, 1% CSC-treated mitochondria were unable to increase Complex-I respiration following the addition of ADP (line 112-113). At 0.1% CSC concentration, ADP induced complex I respiration was reduced by 70% when compared to DMSO control (line 116-117). Why? What is the effect of 0.02 % CSC concentration? You should fully analyze these results in the discussion.
- In mitochondrial oxygen consumption experiment, you pointed out that Isolated skeletal muscle mitochondria were obtained from both male and female adult C57BL6J mice and energized with pyruvate/malate, followed by acute exposures to each chemical and assessment of mitochondrial oxygen consumption (line 167-170). But in CSC treatment on mitochondrial respiratory function experiment, you didn't point out the sex of the animal. Was male or female adult C57BL6J mice used for studying the impact of acute CSC treatment on mitochondrial respiratory function. Why?
- In Figure 4, teal symbols represent male mice, orange symbols represent female mice. Were each individual chemicals detected in male mice and female mice? The quantity and variety were different in Figure 4B-4F.
- Mitochondrial impairments are well documented in skeletal muscle from COPD patients, including elevated mitochondrial reactive oxygen species (ROS). In this study, you did not detect mitochondrial ROS after CSC treatment. Please supplement this experiment.
- The format of references were inconsistent, the doi of several references (reference 1, 2, 3 and the like) was provided, several (reference 10, 14, 20 and the like) was not.
Reviewer 3 Report
Dear editor
Thank you for the invitation to review the manuscript: "NMR spectroscopy identifies chemicals in cigarette smoke condensate that impair skeletal muscle mitochondrial function"
The research design is good and robust. The manuscript is well written, clear and concise.
Small changes are required, namely that all et al. must be in italic once they are in Latin, in all the manuscript.
The introduction is enough to make the context to the reader.
The aim must be written more detailed and clearer.
Results, are detailed and supported by the techniques used.
Figure 4 is hard to read the compounds. Could be improved.
Some references are good and recent, although some are from 2002, 1984, 1988, 1975, 1977, 1994, 1970, just some examples. Which could be replaced with more recent ones.
Strong points the author's acknowledgement of weakened points of the study.
Round 2
Reviewer 1 Report
Dear Editor,
I have confirmed that the manuscript has been corrected and edited by the reviewers' comments as requested.
Thus, it can be acceptable in its present form.
Thanks.
Reviewer 2 Report
The questions have been answered by the author completely.